# Exploring gender differences in HIV-related stigma and social support in a low-resource setting: A qualitative study in the Dominican Republic

**Alane Celeste-Villalvir**[1], **Denise D. Payan**[2], **Gabriela Armenta**[3], **Kartika Palar**[4], **Amarilis Then-Paulino**[5], **Ramón Acevedo**[6], **Maria Altagracia Fulcar**[7], **Kathryn P. Derose**[1,8]*

1 Department of Health Promotion and Policy, University of Massachusetts Amherst, Amherst, Massachusetts, United States of America, 2 Department of Health, Society and Behavior, Program in Public Health, University of California, Irvine, Irvine, California, United States of America, 3 Pardee RAND Graduate School, RAND Corporation, Santa Monica, California, United States of America, 4 Division of HIV, ID and Global Medicine, University of California, San Francisco, San Francisco, California, United States of America, 5 Facultad de Ciencia de la Salud, Universidad Autónoma de Santo Domingo, Santo Domingo, Dominican Republic, 6 Consejo Nacional para el VIH y Sida (CONAVIHSIDA), Santo Domingo, Dominican Republic, 7 World Food Programme, Country Office for the Dominican Republic, Santo Domingo, Dominican Republic, 8 Behavioral and Policy Sciences Department, RAND Corporation, Santa Monica, California, United States of America

* kpderose@umass.edu

**Data Availability Statement:** The data upon which this study is based are in-depth interview transcripts from a small sample of marginalized

## Abstract

HIV-related stigma can affect health by compromising coping and social support. Gender differences in stigma experiences and social support are underexplored, particularly in the Caribbean. We conducted semi-structured interviews (N = 32) with patients at two HIV clinics in the Dominican Republic. Transcripts were coded using qualitative content analysis (deductive and inductive approaches) to identify themes regarding stigma experiences and social support, which were then compared across men and women participants to identify gender differences. While both men and women described experienced stigma, including verbal abuse, men's experience of stigma were subtler and women described outright rejection and instances of physical violence, including intimate partner violence. Both men and women described job discrimination, but women described severe disempowerment as well as permanent loss of income and/or employment whereas men described temporary changes in employment and /or decrease in income. Men and women described modifying behavior due to anticipated stigma, but only women discussed isolating themselves and discomfort taking HIV medication in front of others. Regarding internalized stigma, both men and women described shame, guilt, and depression over their HIV status, though these experiences were more common among women. Women's experiences prevented health care seeking and included suicidality, while men sometimes blamed women for their HIV status and expressed a desire to "move on" and "look ahead." Both men and women described receiving financial support from family and friends, community support from neighbors, governmental support, and support from other people living with HIV. Women most frequently discussed receiving support from family and friends and using religiosity to

and stigmatized people with HIV and contain highly sensitive data that are identifiable via inference. Thus, sharing them would violate the promise of confidentiality made to participants during informed consent. Because the authors did not state in the consent form that the data could be made publicly available to outside researchers, the RAND Human Subjects Protection Committee (HSPC) has stated that the data can not be made available outside the study team. Queries regarding data access can be addressed by the RAND HSPC Chair, Dr. Rebecca Collins (collins@rand.org).

**Funding:** Research reported in this publication was supported by the National Institute of Mental Health of the National Institutes of Health under Award Number R34MH110325, which was awarded to KPD. KP's contributions were supported by a grant from the National Institutes of Health, University of California, San Francisco-Gladstone Institute of Virology & Immunology Center for AIDS Research, P30AI027763 (NIAID); additional funding for KP was provided by K01DK107335 (NIDDK). Revision of the article was supported by R01MH131447, which was awarded to KPD and KP. The article contents are solely the responsibility of the authors and do not represent the official views of the National Institutes of Health. The funders had no role in study design, data collection and analysis, decision to publish, or preparation of the manuscript.

**Competing interests:** The authors have declared that no competing interests exist.

cope, whereas men referenced general family support and government benefits and were less forthcoming about personal relationships and social networks, oftentimes not disclosing HIV status to others. The social context of HIV-related stigma affects women and men differently with physical and mental health impacts and may require distinct mitigation approaches.

## Introduction

HIV-related stigma negatively affects prevention and treatment outcomes across the HIV care continuum. Stigma has been associated with lower rates of HIV testing, as well as denial and lack of disclosure of HIV status, delays in HIV-related treatment, and poor antiretroviral therapy (ART) initiation and adherence [1–3]. HIV stigma is associated with poorer self-reported health status and a higher prevalence of HIV-related symptoms [1]. HIV stigma and discrimination can prevent people living with HIV (PLHIV) from seeking HIV-related health services and disclosing their status to family and friends, resulting in social isolation [4]. Further, stigma undermines HIV-related outcomes by negatively affecting adaptive coping mechanisms and disrupting social support, especially in resource-poor settings [5], while lower HIV-related stigma has been associated with higher levels of HIV resilience (defined as positive adaptation amid hardship) [6].

The HIV Stigma Framework [7] posits that PLHIV experience stigma through three distinct mechanisms (experienced stigma, anticipated stigma, and internalized stigma), and that each process is associated with worse health outcomes and experiences. Experienced stigma refers to the extent to which PLHIV believe they have experienced behaviors that constitute discrimination and other mistreatment by others, whereas anticipated stigma reflects how much PLHIV believe they will experience future discrimination. Internalized stigma comprises negative beliefs and ideas related to HIV that PLHIV internalize [7]. Further, stigma occurs across multiple socioecological dimensions, including the: systemic/structural level (e.g., norms specific to societies, systems, and institutions); community level (e.g., local and cultural norms and values); and interpersonal level (e.g., family, friends, intimate partners, and other close individuals) [8]. Previous studies have found that PLHIV experience stigma and discrimination from family, community members (e.g., friends and neighbors), and healthcare providers in ways that negatively affect retention in HIV care [9].

There is some evidence that being diagnosed with HIV and resultant HIV-related stigma may be experienced differently by women compared to men. Globally, women living with HIV (WLHIV) are more often stigmatized and isolated than men for having HIV, and shamed publicly, name-called, and blamed for bringing HIV into the home or community, even when a male partner is living with HIV [10]. This blame stems from socially accepted norms regarding gender-specific roles, responsibilities, and sexuality in male-dominated societies, where women face inequalities concerning employment, education, and access to healthcare [10, 11]. Additionally, quantitative studies have found that women often score higher on internalized stigma measures in comparison to men in low and middle-income countries such as Ethiopia, Mozambique, Uganda, and China [12, 13].

Gender inequalities and social norms may differentially influence HIV stigma and resilience in the Dominican Republic (DR), another low resource setting in the region of the world (the Caribbean) with the highest adult HIV prevalence outside of sub-Saharan Africa [14]. In 2021, the DR had the highest HIV prevalence of any Spanish-speaking country in LAC (0.9%), [14, 15] and HIV-related complications were the 5th cause of death [16]. The ART coverage rate in the DR was 52% in 2017, and the island-nation adopted a "treatment for all" strategy in

2018 to reduce AIDS deaths and new HIV infections [17]. Nevertheless, it is estimated that only 55% of PLHIV in the DR are on ART, and less than 50% are virally suppressed [15]. Men who have sex with men (MSM), people engaged in sex work, Haitians, and residents of bateyes (settlements run by sugar companies) are among the populations most vulnerable to HIV infection and poor HIV-related outcomes in the DR [18, 19]. Further, the stigmatization of homosexuality and bisexuality by both the Dominican healthcare system and Dominican society at large lead to an underreporting of these behaviors in relation to HIV infection [18]. More than 40% of the population in the DR lives in poverty [20], and the Dominican healthcare system is described as overcrowded and inefficient, with few healthcare providers providing care to PLHIV that is free from discrimination [18]. Women represent more than half of the adults living with HIV in the DR, in comparison to Latin America where men are overrepresented in the proportion of PLHIV [21]. A study with PLHIV engaged in sex work in the DR and Tanzania found many women had not achieved viral suppression, had high rates of HIV drug resistance, and suboptimal ART adherence [22].

In terms of HIV stigma in the DR, previous studies have found that WLHIV frequently face internalized stigma and depression [23]. A qualitative study exploring the impact of HIV diagnosis among WLHIV in the DR found that stigma and lack of disclosure hindered HIV self-management and ART adherence, as some women avoided HIV clinics out of fear of being seen there and having her HIV status disclosed [24]. Another DR-based study with key stigmatized populations (female sex workers, men who have sex with men, and people of Haitian descent) found the highest levels of experienced stigma among female sex workers (FSWs), including verbal and physical assault and harassment—this dynamic was not present among MSM, many of whom had higher socioeconomic status than FSWs [25]. This study associated higher levels of stigma with higher odds of missing an ART dose [25]. A study evaluating barriers to viral suppression among PLHIV in the DR found that male and employed patients were more likely not to be virally suppressed, partially due to stigma that limited engagement in care [26].

Along with stigma, low social support has been widely recognized as a factor adversely affecting engagement in HIV care [26]. In a DR-based study with WLHIV, participants described controlling disclosure, educating self and others about HIV, and seeking support from family, friends, or others [27]. In a study about HIV management with FSWs living with HIV in the DR, participants mentioned fear of stigma and lack of social support as factors that negatively impacted HIV management, while support from family and friends improved self-esteem and confidence to management HIV and cope with stigma [28].

To our knowledge, prior studies have not explored differences in HIV-related stigma experiences and social support between men and women living with HIV in the DR. Many studies on HIV-related stigma in lower- and middle-income countries are focused on internalized stigma, creating a research gap about the behavioral and treatment-related impacts of anticipated stigma and experienced stigma [1]. Previous studies examining HIV stigma experiences and social support among PLHIV in the DR have focused on specific populations, such as women, MSM, or FSWs [29–32]. The objective of this study is to explore and compare experiences of HIV-related stigma (experienced, anticipated, and internalized) as well as social support and coping mechanisms by gender among PLHIV more generally in the DR.

## Methods

### Ethics statement

All study procedures were approved by the Institutional Review Boards at the RAND Corporation, Universidad Autónoma de Santo Domingo (UASD), and the Dominican Ministry of

Public Health. Verbal informed consent was obtained from all study participants, since the principal risk would be potential harm resulting from a breach of confidentiality if written consent were used. Each interviewer gave participants a written consent statement and reviewed each section orally, answering any questions during the consent process. Names and other identifying information were not included in interview transcripts to protect participants' confidentiality.

## Author positionality

The authors represent diverse, intersectional identities and have relevant professional and lived experiences. All authors are fluent and literate in Spanish; six are from Latin America (four from the DR) and the other two have many years of experience living and working in Latin America. The team included diverse professional backgrounds (public health, health policy, medicine, and nutrition) and individuals who are active in the networks of PLHIV and the LGBTQ community in the DR. Six of the authors have doctoral degrees and were working as researchers at the time of the study; the other two have college degrees and were working in the areas of food security and HIV. Seven of the authors are women and one is a man.

## Study overview

This qualitative study comes from a formative research component of a larger community-partnered study conducted by RAND and UASD, in collaboration with other Dominican partners: The Ministries of Agriculture and Public Health, the Dominican National HIV/AIDS Council (CONAVIHSIDA), and the World Food Programme. The aim of the larger study was to develop and pilot test an integrated urban gardens and peer nutritional counseling intervention with PLHIV and food insecurity to improve ART adherence and treatment outcomes. This intervention was based on prior research on food insecurity among people with HIV in the DR that was co-led by RAND and WFP in partnership with CONAVIHSIDA. The formative research component, conducted before the pilot cluster randomized controlled trial (RCT), involved qualitative interviews with patients at two study clinics to explore food insecurity, nutritional knowledge and behaviors, ART adherence, and other factors influencing HIV outcomes to inform the development of the integrated intervention [33]. Community partners (community leaders from CONAVIHSIDA, WFP, the Ministries of Agriculture and Public Health) were involved in all phases of the research, including development of the research questions and interview guides, review of preliminary findings, and dissemination.

The present study uses qualitative content analysis [34] to explore how PLHIV in the Dominican Republic experienced, anticipated, and internalized stigma and social support and how these experiences may have differed by gender. Qualitative content analysis was selected to address this study question due to the flexibility of utilizing multiple approaches (e.g., inductive and deductive) in order to derive meaning and to understand the phenomena in question [35]. Further, content analysis is a method that is well studied and provides an effective means of organizing and deriving meaning and conclusions from textual data [35, 36].

## Sampling and recruitment

For the purposes of the cluster RCT, including the initial formative research, the study clinics were selected purposively to both be urban, government-operated clinics in central and northwestern DR of similar size and staff composition with comparable standards of care. The clinics were in provinces with the country's highest HIV prevalence but not in provinces where our team had done prior pilot gardens. The clinics were in similar regions but far enough to

avoid cross-contamination. The study team recruited PLHIV with moderate or severe food insecurity from two study clinics to participate in semi-structured interviews.

Adult patients were approached on the day of their clinic visits and screened for eligibility by two Dominican research assistants using a structured questionnaire. Eligibility criteria included: 1) being a registered patient at the HIV clinic, 2) residing in an urban or semi-urban area, 3) being 18 years and older, and 4) having moderate or severe household food insecurity according to the Escala Latinoamericana y Caribeña de Seguridad Alimentaria (ELCSA; α = .91-.96 across LAC countries) [37, 38]. All participants were asked to self-identify their gender and all indicated "man" or "woman." The study team purposefully recruited a sample that had approximately equally numbers of men and women and a range of ART adherence (i.e., self-reported adherence problems versus not). Participants who met eligibility criteria were provided additional information about the qualitative study and, if interested, were invited to complete a one-time interview.

## Data collection

Thirty-two interviews were completed between May-September 2017 by two trained local interviewers and co-authors (RA, MAF), both of whom had extensive experience with PLHIV, college-level education, and training and experience with qualitative interviews. All interviews were conducted in private rooms and in Spanish, lasted between 60–90 minutes, and were audio recorded with permission. Interviewers took field notes during the interviews. The semi-structured interview guide was adapted using a previous guide developed by the researchers [24, 39] and with input from local partners. Because the purpose of the interviews was to inform the development of the nutritional counseling and urban gardens intervention for people with adherence difficulties, the guide focused on topics related to nutritional knowledge and beliefs, food acquisition and dietary behaviors as well as economic and food security, access to health services, experiences with HIV, and ART adherence. Underneath the heading of experiences with HIV, participants were asked about their experiences with diagnosis, physical and mental health changes, social relationship changes, social support, feelings of guilt or shame, and whether they have been victims of verbal or physical violence. Thus, the guide approached questions about "stigma and discrimination" indirectly, by asking questions to elicit information about the nature of participants' interactions with the various levels at which stigma and discrimination operate (institutional, community, interpersonal, and intrapersonal) (see Fig 1 for levels and examples of questions). Socio-demographic data were collected using a structured questionnaire with questions about age, gender [man, woman, or other (specify)], nationality (specify), educational attainment, primary occupation, marital status, household composition, and monthly household income. Interviews were conducted until thematic saturation was reached and no new overarching topics or themes emerged in the data. Audio recordings were transcribed verbatim by a Dominican research assistant and verified by team members. Using the Consolidated Criteria for Reporting Qualitative Research (COREQ) guidelines, steps were taken to ensure the trustworthiness of the process of data collection and resulting data, including the collaborative development of instruments, providing training to interviewers, the use of field notes and an interview guide during interviews, and audio-recording [40].

## Data analysis

Coding procedures for the full dataset (i.e., across all topics, not just stigma experiences, coping, and social support) consisted of qualitative content analysis methods [34] using deductive approaches while also allowing for emergent codes [41, 42] or inductive approaches.

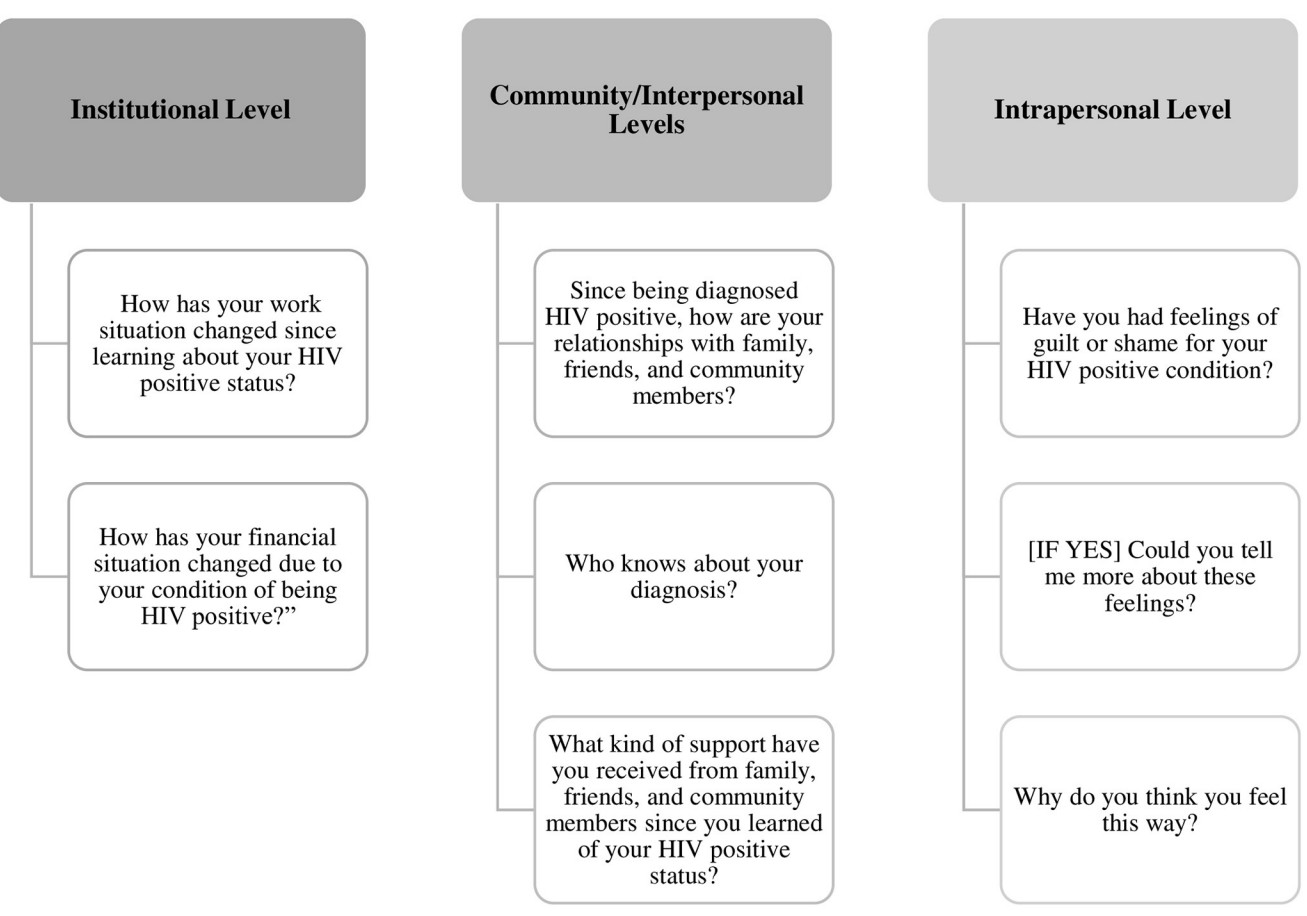

**Fig 1. Examples of interview guide questions by socioecological level.**

Transcripts (in Spanish) were uploaded to Dedoose [43], a web-based text management and analysis software. The original, Spanish-language transcripts were used since all team members were fluent or native Spanish-speakers. Two coders, one with a PhD and extensive experience with qualitative methods and the other in a PhD program being trained on qualitative methods, read and coded transcripts independently and used a multi-stage exploratory approach to develop a codebook corresponding to the main topics of interest in the interview guide. This approach involved creating initial codes based on the interview guide, coding the data, then grouping similar codes to create categories [44, 45]. In re-reading transcripts, coders wrote memos for text that did not adhere to the established coding structure, which were used to generate emergent codes. Qualitative content analysis has been identified as "unique" in being able to combine deductive and inductive approaches [35]. Of note for this analysis, coding was done without considering an individual's gender. Any discrepancies in coding were resolved by consensus. Trustworthiness in this initial phase of data analysis was facilitated through the using memos during the coding process, developing an iterative codebook, engaging multiple independent coders, and resolving discrepancies via consensus [40, 46].

For the current study on gender differences in stigma, coping, and social support, the codes of interest focused on participants' experiences with HIV stigma and specifically as they related to: 1) experiences with being diagnosed with HIV; 2) changes in physical health after diagnosis; 3) changes in social relationships after diagnosis; 4) HIV disclosure to others; 5) experienced

and anticipated stigma; 6) internalized stigma; 7) social support; and 8) stigma–other. In addition, we examined narratives coded under: 9) their current work status and 10) changes in income or work due to their HIV status. For each of these 10 codes, we extracted code reports from Dedoose [43] by gender, where all participants' narratives coded to that code were listed. Then two authors, another with a PhD and extensive experience with qualitative methods and the author in a PhD program being trained in qualitative methods, read through each code report to develop analytical memos describing prominent themes related to stigma experiences, coping, and social support. These memos were documented in matrices organized by participant and code, with one matrix for men and the other for women, which allowed for constant comparison of patterns–including salience, range, convergence and divergence–by gender across participants and within codes [47]. The qualitative data, at all points of this analysis, were examined for semantic (also known as manifest) meaning, without regard for substratum and latent meaning that can be interpreted in participant physical gestures, laughter, sighs, and silence, as these can oftentimes result in an over-interpretation of data [35, 46]. The focus instead was on participants' narratives tagged as relating to one of the codes relevant to the research questions [48].

The matrices were reviewed and discussed by the qualitive analysis team (all bilingual and doctoral-level trained in qualitative methods) to identify preliminary themes regarding gender differences in stigma experiences and social support, which were then discussed among co-authors and other partners in the DR for feedback. Once themes were finalized, exemplary quotations were selected and translated to English by bilingual and Dominican lead author and reviewed by other bilingual authors for inclusion in this paper. Themes were arranged using constructs adapted from the HIV Stigma Framework [7] and presented in terms of their salience, contribution to range, and how they converged or diverged between men and women. Trustworthiness in this phase was facilitated by using analytical memos, extensive discussion and constant feedback as a team (including with team members with lived experience), developing matrices with the themes identified in the code reports to further organize the data, and using representative quotations to illustrate themes and trends in the data [40, 46].

## Results

### Participant characteristics

Thirty-two PLHIV participated in this study (Clinic 1 = 18; Clinic 2 = 14), whose characteristics–collected by structured questionnaire–are presented in Table 1. Participants had a mean age of 45 years (range: 26–72 years). The sample was purposively split equally between men and women (n = 16 each). Most women and men had a primary school education or less. Most men (n = 10) and women (n = 9) were not legally married but living with a steady partner. More women (n = 13) than men (n = 9) reported experiencing severe food insecurity as indicated from their responses to the ELCSA. As a result of our purposive sampling, half of each gender group reported low ART adherence (i.e., took ≤ 75% of their ART medications in the past month) (Table 1).

In terms of primary occupation, most women (n = 11) reported being unpaid homemakers while the remainder primarily worked informal jobs (e.g., providing childcare, selling clothing, as street food vendors, health aides, and domestic workers). Men tended to be employed (5 full-time, 3 part-time, 5 self-employed) in paid labor outside the household. Men primarily performed informal, potentially hazardous jobs requiring manual labor, including construction, painting homes and/or fences, and agricultural labor. Other male participants described work as clothing or food vendors in local street markets. About a third of participants overall self-reported a monthly household income of <5,000 Dominican pesos, the equivalent of

**Table 1. Participant demographics across two Dominican HIV clinics (n = 32).**

| | Total (n or mean) | Women (n = 16) | Men (n = 16) |
|---|---|---|---|
| Age (mean, SD) | Mean: 45; SD:10.5 | 42 | 48 |
| Gender (self-reported) | 32 | 16 | 16 |
| **Nationality** | | | |
| Dominican | 27 | 12 | 15 |
| Haitian | 3 | 2 | 1 |
| Unknown | 2 | 2 | 0 |
| **Education** | | | |
| Primary | 20 | 11 | 9 |
| Secondary | 7 | 4 | 3 |
| None | 5 | 1 | 4 |
| **Primary Occupation** | | | |
| Employed (full-time) | 5 | 0 | 5 |
| Employed (part-time) | 4 | 1 | 3 |
| Homemaker | 11 | 11 | 0 |
| Self-employed | 8 | 3 | 5 |
| Unemployed | 1 | 0 | 1 |
| Other | 3 | 1 | 2 |
| **Marital Status** | | | |
| Legally married and living together | 1 | 1 | 0 |
| Living with a partner, not legally married | 19 | 9 | 10 |
| Separated/Divorced | 12 | 6 | 6 |
| **Monthly Household Income (Dominican Pesos)** | | | |
| <5,000 (~$100 USD) | 14 | 7 | 7 |
| 5,000–9,999 (~$100–200 USD) | 10 | 4 | 6 |
| 10,000–19,999 (~$200–400 USD) | 4 | 2 | 2 |
| 20,000 or more (~$400 USD) | 2 | 1 | 1 |
| Missing | 2 | 2 | 0 |
| **Food Insecurity Status*** | | | |
| Moderate | 10 | 3 | 7 |
| Severe | 22 | 13 | 9 |
| **ART Adherence Status** | | | |
| Adherence difficulties | 18 | 8 | 10 |
| No adherence difficulties | 14 | 8 | 6 |

*Food insecurity measured using the Latin American and Caribbean Food Security Scale (ELCSA) [37]

about US$100 per month, or US$3.33 per day [the lower middle-income class poverty line is $3.20/day per capita [49]]. Most (22 of 32) participants had "severe" food insecurity as defined by the Latin American and Caribbean Food Security Scale (ELCSA), since having moderate or severe food insecurity was an eligibility criterion for the overall study; among our participants, women had higher rates of severe vs. moderate food insecurity than men.

## Experienced stigma

**Interpersonal and community-level stigma.** In the qualitative interviews, women and men participants discussed at length the different types of stigmas they encountered since being diagnosed with HIV, including experienced stigma. Verbal abuse and discrimination

from others was a salient theme among both men and women, such as people avoiding physical contact with them out of fear of acquiring HIV and people using the term "sidosa/o" to refer to them—a pejorative slur to refer to someone with HIV that categorizes a person with HIV as having AIDS. A woman described an interaction in which she was referred to as "sidosa" as follows,

*I have received support from neighbors, just like I have received rejection as well. They say to me, "look at this sidosa coming" or "be careful with that one, she has AIDS" and sometimes I have to act like I don't hear them because if I pay attention to this then I will really die quickly*

*(ID 101, 36-year-old woman).*

A man described a similar experience during which this term was used to insult him,

*. . .this 'sidoso' she told me. . .that woman yelled at me, called me 'sidoso' and every-thing. . .those are things that cause me shame, you know. . .even though I know what I have, but no. . .nobody needs to be yelling those things. . .*

*(ID 215, 50-year-old man).*

This term was used in face-to-face interactions, or used by neighbors, friends, family to gossip and badmouth participants to others.

Although both women and men described experience with verbal abuse and discrimination due to their HIV status, there were some differences in the qualitative nature of these experiences, as described by the women and men. For example, both women and men described others avoiding casual contact with them in public. For example, one man explained *"it was such that, a cup that you drank out of. . .there were people that didn't drink from it. . .and if you sat in a chair, there were also people who didn't sit there (ID 208, 61-year-old man)."* But women's descriptions tended to emphasize rejection by their closest family and friends, and women provided detailed descriptions and examples of how they were shunned, rejected, or ostracized. For example, here one woman describes rejection from her extended family,

*Look, I have a sister whose house I don't visit. I don't visit her house because her husband thinks that this [HIV] is like a cold, that you sneeze and he can catch it. Sometimes, when you have a cold, you sneeze and the germs can pass on the cold to others, and they think it is like that. To avoid problems, I just don't go near that person, I don't give them anything to eat in my house, I don't even give them water. . .nothing*

*(ID 112, 52-year-old woman).*

Another stark difference between women and men participants' stigma experiences was that women disclosed specific incidents of physical violence, intimate partner violence, and sexual assault in response to their diagnosis, whereas men did not. For example, one woman described how she was physically (i.e., slapped, pushed) and verbally abused by friends due to social judgement and ignorance about transmission,

*Because you know that when you have this condition, people don't accept you the same, but instead think that you're a licentious person, that you're not worthwhile, like they say. . .. They think maybe that you will get them sick or cause them harm. It [HIV] is something they do not understand. Since they don't know about it [HIV], they mistreat us. I had this need to be close to others, to be friends with them, but they rejected me and I tried to be involved with*

*them [friends]. They pushed me. . .Many of them slapped me. They slapped me. . .yes, they would say to me, "sidosa, get away from here. . ..*

*(ID 101, 36-year-old woman)"*

Another woman described the physical violence she experienced from her significant other–violence that ended in the tragic loss of her child–coupled with perceived isolation from her family,

*He mistreated me a lot. . .because of my condition and because my family did not love me, because he'd say that if a family doesn't love you, you're worse than a dog. He didn't care about hitting me or mistreating me because, when he hit me, who in my family would come? Nobody. In one instance, he wanted to kill me with a gun, and I went to look for a restraining order and I was able to escape his mistreatment. . .I said thank God. That was years ago. . .he caused the death of my 3-month-old newborn, because with my 7-month pregnancy, he beat me up and I had to have a c-section. The baby's lungs would not mature. After I got a restraining order, he has not ever been a man to me. I have years without seeing him*

*(ID 109, 38-year-old woman).*

While both men and women faced rejection and verbal abuse, women described experiences of physical abuse and violence requiring criminal justice involvement.

**Employment discrimination.** Men and women participants both described experiences with job discrimination because of their HIV status. Men most often worked informal jobs such as construction and painting. Men participants discussed past experiences with employment discrimination, including employers who were hesitant or resistant to hire them due to their beliefs that they posed a risk of transmission. Most men also described a loss of/or change in employment status and/or income after being diagnosed with HIV, as described by this man, "*Yes, I had my job. . .they fired me. . .I spent 13 years working in that company and, as soon as they found out I had HIV, they fired me (ID 211, 48-year-old man).*" Another man indicated that his employability had changed since prospective employers feared what might happen if he accidently cut himself on the job (thinking that his blood would be a risk for transmission),

*There has been change, perhaps, because whoever wants to pay me for a job says, "well, that man has HIV, if I give him a job and he cuts himself, now we have a problem" and so on and so forth, because there are people who believe that even with sweat [HIV is transmitted] . . .you see?*

*(ID 205, 59-year-old man)*

Nearly half of the women reported losing jobs and/or income, and most formerly worked in restaurants and in homes (cleaning or providing childcare). One woman described how she lost her job at a restaurant right after being diagnosed with HIV, "*Primarily when I came out positive for HIV, I think I was 15 years old, to be honest. I worked as a waitress in a restaurant. Immediately after they knew my test results were positive, they fired me. That was a very difficult experience for me (ID 217, 32-year-old woman).*"

A distinction between men and women, however, was that experiences with rejection were central only to the narratives of women participants. As an example of this, here one woman describes how she lost customers and had to close her business due to discrimination and rejection,

*I used to have a salon, and because of my condition, people really like to speculate and talk, and there are a lot of people who discriminate. . .I closed it [the salon], I left the salon. . .it affected me because I used to earn well. I used to earn good money in the salon, I did pedicures, manicures, hair, everything. But I closed it for that reason because I saw people distance themselves. . ... My source of income changed, that salon was everything for me*

*(ID 112, 52-year-old woman).*

Compared to the severe disempowerment and altogether loss of income and employment experienced by many of the women participants, men more often described *temporary* changes in employment, such as a loss of the employment they held at the time of diagnosis or a decrease in income due to lower community patronage. As described by a participant, "*. . .When I was in my normal state, or what I thought was normal. . .I had more work. . .My wife would call me to paint a gate. . .I would paint and earn more money. I have less people with this [HIV]. . ... yes, my income has decreased*" (ID 219, 43-year-old man). In contrast, feelings of rejection as a result of job loss were more frequently articulated by women participants and were more emotionally charged. For example, one woman participant explained, "*. . .there are people who discriminate because, look, one time I was taking care of a child and when it was discovered [HIV diagnosis] . . .they stopped bringing the child*" (ID 112, 52-year-old woman).

## Anticipated stigma

Both women and men participants anticipated job-related discrimination, such as being terminated from jobs after people at work discovered their HIV status, and reported not seeking employment if they believed they might be tested for HIV. Unemployed women and women engaged in informal work frequently described being hesitant or afraid to seek formal employment because they anticipated employment discrimination (e.g., mandatory blood test). One woman described this fear of employment discrimination, "*I can't go to a company and say that I want a job, or 'give me a job', out of fear that they will test me or maybe reject me*" (ID 105, 34-year-old woman). Similarly, men mentioned that they avoided seeking certain jobs so that they would not be tested and potentially shamed for having HIV, as described by this man,

*If you go to an area to work, you are afraid of working [there] because, as soon as you get there, they send you to get tested. And sometimes you limit yourself. What do I do instead? I don't go to work at those companies because of the shame, because as soon as you are there, they send you to get tested*

*(ID 219, 43-year-old man).*

Men and women participants described distancing themselves or modifying their daily behavior out of fear of discrimination or rejection, as one participant explained, "*you know I don't eat in other people's homes. I can eat in the homes of my family, but in other people's homes. . .I don't dirty up plates because there are many who say, 'oh, but so and so has. . ..' They change my name and call me 'HIV' . . ..*" (ID 211, 48-year-old man). Women also mentioned being reluctant to physically interact with others in case the person was uncomfortable with PLHIV, and sometimes even with individuals who expressed comfort with PLHIV. As described by one woman participant, "*No, if I see that a person discriminates, I simply distance myself from that person and stop paying them any mind because that is a person who is closed-*

*minded and does not understand that this [HIV] is not a sickness but rather a lifelong con-dition. . ." (ID 201, 28-year-old woman).*

Women's and men's narratives diverged on this issue in that women sometimes discussed fears of being emotionally rejected and reacted by isolating themselves from others to prevent rejection. For example, one woman described deliberately isolating herself from others by saying, "[*Do I feel] guilt? No. But, for example, sometimes I get away from people. I feel weird about being rejected, and before they can reject me, I get away (ID 201, 28-year-old woman).*"

Women participants described other behaviors that demonstrated anticipated stigma and may impact ART adherence, including hiding what their ARTs were for, discomfort in taking medication in front of others. Regarding taking her ART medications in front of others, one woman stated that to avoid disclosing her HIV status, she tells others, "'*Well, I have a bit of a cold,' because sometimes I don't want them to know because there is so much rejection that I don't want them to know anymore, instead sometimes I have to tell pious lies (ID 101, 36-year-old woman).*" Men did not describe this same phenomenon of anticipated stigma affecting their ART behaviors, though they did more often mention that they withheld their HIV status even from close family and friends. One participant noted that, "*My children do not know. . .I keep it confidential. . .the only people who know are me and my wife (ID 213, 58-year-old man).*

## Internalized stigma

Both men and women experienced feelings of shame, guilt, and depression over their diagnosis and HIV status. For example, here one woman described feelings of disbelief, "*I didn't believe it; I even went to [name of another city] to do it [HIV test]. I did it [HIV test] like 3 times, and it repeatedly came up positive*" (ID 104, 58-year-old woman). Similarly, a man participant described feelings of guilt,

> *Sometimes I start to think, I feel guilty, because the only one who is to blame has been me. . .but I keep pushing forward, I keep going, because if I get sick because of that, I can stop working. . .. you have to keep pushing forward.*
>
> *(ID 114, 50-year-old man)*

The way women and men's narratives diverged on this point was that most women we interviewed highlighted feelings of shame, guilt, disbelief, suicidality, and depression, whereas few men expressed these same feelings. Women's reactions to being diagnosed reflected intense sentiments that could impede health care seeking behavior. For example, here one woman describes her disbelief and difficulty acknowledging the diagnosis, stating, "*I'll tell you that it took me, it took me a long time to be able to accept it (ID 112, 52-year-old woman).*" Another woman described shame so intense that it prevented her from seeking HIV treatment, "*Well, when I found out [HIV diagnosis], I couldn't believe it because it was the father of my daughter. I got very skinny. . .I spent four years without coming here [to the HIV clinic] because the shame was killing me (ID 106, 41-year-old woman).*"

Other feelings highlighted by women included low self-esteem, sadness, powerlessness, lacking strength, anger, a desire for revenge, and feeling worthless. For example, here one woman describes feeling impotent, engaging in substance use, having low self-esteem, and spending an extended period of time not taking ARTs,

*When people say obscene or ugly words to me, my self-esteem wants to go lower. I feel impotent when I feel that people speak badly of me. . .Because, I have had low self-esteem because of the discrimination, because I didn't have a job. . .I fell into drugs and lasted a long time without taking medication [ART]*

*(ID 101, 36-year-old woman).*

Other women similarly described feelings of anguish and periods of severe depression after their diagnosis, attributing it to the belief that death was imminent and that they had nothing to lose. One woman's experience led to weight loss, depression, and lack of sleep out of fear of death, as she describes here:

*Imagine, so many years with one person, and I had no idea I had this [HIV]. I was so innocent, I thought that it [HIV] was something in the water. . .but when I learned it was through sexual relations, I went out screaming, and they had to restrain me. I got depressed and didn't sleep or anything. I weighed 120 lbs. . .I had heard people talk about it [HIV], that whoever had that [HIV] would be dead by morning. . .so I didn't sleep.*

*(ID 104, 58-year-old woman)*

Another woman described a period following her diagnosis of severe emotional crisis, fears of hurting her daughter, social isolation, and substance use because she believed the diagnosis was a death sentence,

*When I got there, I locked myself in and started crying. The window was a bit high for me, but I wanted to throw my daughter, kill my own daughter. My mother took her away. . .I didn't eat, I didn't do anything. I lived shut in. . .. People who I knew who I told, didn't believe me. . .they'd get away from me. . .I would go to sleep drinking, smoking, I was destroying myself. I was destroying myself. I would say, well. . .since I am going to die*

*(ID 109, 38-year-old woman).*

In contrast, men participants did not frequently describe feeling depressed, suicidality, shame, or guilt. Rather than describe experiences that reflected internalizing or being impacted emotionally by their HIV diagnosis, men participants tended to externalize and blame women for them contracting HIV, as suggested by this man's comments,

*I didn't feel guilt toward anyone because, if I caught this, I caught it because I was out in the street. . .but you don't only catch this in the street, you can catch it in your own home because you say, "Oh, I take care of myself, I don't go out to the street" but your wife does. Your wife can go out and find a man who she likes more who will give it to her without a condom, and then from that other guy she comes and brings it to you. . . Not [only] HIV, no, any venereal disease that you can catch 9ID 205, 59-year-old man).*

Men participants more often discussed how they were "shy" (i.e., not divulging/disclosing status) or "calmed down" (i.e., from a previously promiscuous lifestyle). One participant explained his change in behavior, *No [feeling shame] because I have been a person that, immediately as this [HIV] came into my body, I don't take risks in terms of having relations with a person. . .if I don't have a condom, it is better to let the moment pass because moments are moments and life is life. . . (ID 113, 47-year-old man).*

**Table 2. The impacts of different types of stigma on women and men living with HIV in the Dominican Republic.**

| Stigma Mechanism | Women | | Men | |
| --- | --- | --- | --- | --- |
| | Examples | Consequences | Examples | Consequences |
| **Experienced Stigma** | 1. Verbal abuse and discrimination<br>2. Job discrimination and loss<br>3. Blatant rejection from community, family, and friends<br>4. Physical violence, intimate partner violence, sexual assault | • Avoid people and physical contact<br>• Lost job, closed business (e.g., beauty salon)<br>• Do not visit family; avoid serving food or drink to others in one's home<br>• Lost pregnancy | 1. Verbal abuse and discrimination<br>2. Job discrimination and loss<br>3. Subtle discrimination (avoiding casual contact) | • Avoid physical contact<br>• Lost job |
| **Anticipated Stigma** | 1. Fears of rejection<br>2. Fears of employment discrimination | • Social isolation<br>• Avoid jobs that require medical testing<br>• Will not take ART in public; will conceal what medications are for | 1. Fear of employment discrimination<br>2. Fear of rejection | • Avoid jobs that require medical testing<br>• Do not disclose status; do not eat in others' homes |
| **Internalized Stigma** | Feelings of shame, guilt because of HIV+ | • Suicidal ideation, depression,<br>• Substance use, ART non-adherence, avoidance of HIV clinic | Some with shame initially but "moved on" | Changed lifestyle, "calmed down" |

Some men mentioned feeling initial shame about HIV status when diagnosed, but not at the time of the interview. Men commonly used optimistic terms and perspectives—of "moving on" and "looking ahead" as a coping strategy. As one man explained his shift in perspective, "*at first, I had a lot of shame in the beginning, but not anymore. . .because there are a lot of us [with HIV] and people have to open their minds because if we have our minds closed then we die sooner (ID 216, 50-year-old man).*"

It was common for men to not express any feelings concerning their HIV status during an interview–their responses were comparatively brief and did not use emotional language relative to women's comments. For instance, when asked whether they experienced shame or guilt concerning their HIV status, several men responded with a brief, "no", even when probed for more detail. Others elaborated a bit more, like this man, who stated, "*No, I didn't pay any mind to that [having shame/guilt] . . .I didn't pay mind to anything (ID 207, 50-year-old man).*" Similarly, another male participant described having a good state of mind, "*No, thank God, no. My mind is always good (ID 212,37-year-old man).*"

To summarize the themes regarding experienced. anticipated, and internalized stigma, Table 2 provides an overview, comparing examples and consequences of these different types of stigma between men and women participants.

## Social support

Both men and women described having access to social support and other sources of resilience that helped them cope with HIV-related challenges and marginalization. These supports included social support from immediate family and friends, community support from neighbors and church networks, institutional and governmental support, and the support of other individuals living with HIV. For example, here one man described moral support from family, specifically because not many other people knew of his diagnosis, "*Yes, my family gives me advice so that I am not sad. . .you know, how do I tell you. There are people who do not know that I am sick, they do not know that I suffer*" (ID 102, 38-year-old man). An example from a woman participant described material support from her daughter, "*My daughter [provides support] . . .she sends me cooked food, every day except Sunday. . .*" (ID 103, 73-year-old woman).

When we compared the amount and sources of support between men and women participants, we saw some differences. For example, several women participants described how their daughters provided support, including adult daughters who no longer resided with them. A woman described moral support from both her mother and daughter, "*My mother. . .I am with her almost every day. . .but also with my daughter, she has come here [to the HIV clinic] to the program as well. . .sometimes, when she is able, she has come with me*" (ID 202, 37-year-old woman). Women participants most frequently described emotional and moral support but some mentioned material resources (i.e., food) or reminders concerning HIV care (i.e., taking medication or attending medical appointments). For example, one woman explained how her neighbor supports her with food, "*My neighbor just tells me, 'Did you eat yet?' if I have not eaten, he says no and gives me food*" (ID 110, 30-year-old woman). Similarly, another woman described her despair upon learning of her diagnosis and how her network of friends provided social and economic support so that she could buy food,

> *I came here to the hospital and they told me, "You have AIDS." I thought of taking poison and poisoning myself, of taking a rope and hanging myself. . .I thought a lot of things, but what can I say? I always have a lot of friends. My house was full of people, and when they would see me crying, they would say "[Name redacted,] what is the matter? [Name redacted], are you hungry? Here," and they would get together and gather food, money. I was able to collect about 5,000 pesos, with people giving me money so that I did not go hungry*

> *(ID 204, 57-year-old woman).*

Similar to women participants, men noted moral and financial support from family and friends. Men highlighted that friends and family accompanied them to medical appointments, provided money for transport for medical appointments and to get ART, money for bills, and money and food were also sent to them from overseas (e.g., canned food from relatives in the U.S.). For example, here one man described how friends help him with gas money so he can get his medications, "*I have a little beat-up car over there, and I ask certain friends, 'hey let me get 50 pesos so I can put gas so I can go to the hospital to get my medications'. . .I thank God that nobody says no*" (ID 205, 59-year-old man).

Men also mentioned emotional and moral support from friends at the HIV clinic, an experience that did not come across in the narratives of the women. For example, here one man describes how the support he receives at the clinic helps improve his self-esteem,

> *. . .the support I get from family is not the same as when I come here [to the HIV clinic] . . .I disconnect from civilization and work, and when I get out of here, I come out focused with a different vision. I come here to recharge, specifically. When I come here, it's like my self-esteem improves.*

> *(ID 108, 43-year-old man)*

Several women mentioned benefitting from the Solidaridad program, a monthly food assistance program that provides recipients with $825 Dominican Pesos for groceries (approximately $15 USD per month). For example, here one woman described how she is able to support her family by accessing this public benefit, "*Yes, I have the Solidaridad card, and with that and what I earn when they call me sometimes [to do informal domestic work], I support my three children, my husband, my grandfather. . .. (ID 105, 34-year-old woman).*"

Women participants also discussed spirituality and religiosity as forces of strength and resilience. For example, this woman described how faith lifted her out of a dark period following employment discrimination and rejection,

> *I was without work because people living with HIV are mistreated. The people who have businesses do not want these types of people [HIV positive]. For example, I was working at a local place where food was sold, and a lot of people. . .would stop buying food there because of my condition. . .Until the man from the business kicked me out, told me to leave the business because they [customers] didn't want to buy his food. And I felt so rejected, but Christ came and picked me up, glory to God*

> *(ID 101, 36-year-old woman).*

Unlike women, men were less forthcoming about their personal relationships and social networks, and some shared they did not disclose their HIV status, in some cases withholding their status from their family and close friends. As one man described, *"No, no, no, no. . .my Friends, nobody knows anything about that [HIV status]. My friends, I have not told any of them"* (ID 111, 45-year-old man).

Men participants mentioned obtaining material support from local and government institutions (e.g., food bags from the HIV clinic and food from a visiting political candidate). A few mentioned other types of governmental support, such as access to subsidized health insurance, free or low-cost medications, and access to the aforementioned Solidaridad food assistance cash card. For example, here one participant described several types of assistance,

> *One of these days, my wife came here [to the HIV clinic] and they gave her a little bag [of food] . . .. that is one day out of the year. . .I suppose they should give it to us weekly, monthly, I think. . .. we are the ones who need it. Sometimes my mother had a little card [Solidaridad], we received $800 pesos monthly [~$15 USD] . . .but imagine, 3 people eating per month*

> (ID 219, 43-year-old man).

Other participants, while somewhat familiar with Solidaridad, were unfamiliar with the program's eligibility criteria and did not know where or how to apply but wished to have such monthly food assistance.

## Discussion

This qualitative study with PLHIV in the DR reveals the various types of stigma (experienced, anticipated, and internalized stigma) that both men and women experience as PLHIV. While these types of stigma have been found to cause delays in HIV treatment, poor ART initiation and adherence, cause social isolation, result in missed ART dose, and lower HIV resilience [1, 4, 6, 25], it is not well known how these different types of stigma may be differentially experienced by gender. Our study found several differences between men and women in their experiences of HIV-related stigma with important implications for addressing this pervasive and persistent barrier to optimal HIV outcomes.

In terms of experienced stigma, both men and women said others avoided physical contact with them and verbally abused them, which are experiences previously documented in the literature [4, 9]. Our study found these experiences were mentioned more frequently among women, who emphasized rejection from family and friends and being the subject of gossip. While men made note of discrimination, they perceived it as less blatant and nonviolent (e.g., people avoiding contact with items they had touched). Women, on the other hand, described

physical violence, intimate partner violence, and sexual assault. Our study is consistent with a review of qualitative studies with WLHIV across low and high resource settings that identified physical violence as a common experience among women [11]. High rates of intimate partner violence (IPV) have been documented in the DR in the general population with recent increases in physical IPV in contrast to declines in other countries in the Americas [50], suggesting that underlying attitudes regarding violence towards women should be addressed. Gender violence increases women's vulnerability to HIV in the DR and is driven by poverty, discrimination, and machismo [18]. Previous studies demonstrate that gender-based violence limits women's access to HIV testing and treatment [51]. Our study further underscores the intersectional nature of stigma, suggesting that integrated responses to gender-based violence and HIV are needed, particularly those that embed an intersectional stigma framework that responds to the distinct yet interconnected forms of stigma and discrimination WLHIV are subjected to due to concurrent marginalization along axes of class, gender, and HIV status [52]. A recent study in the DR describes such a model, with gender-based violence trainings for service providers and peer educators in an HIV clinic [53].

Among the most prevalent forms of experienced stigma described by participants was job discrimination. Men and women described firsthand experiences with job discrimination, including job loss and/or changes in income and employment status post-diagnosis as well as a decline in employability due to employer fears of contracting HIV through common work tasks. Such experiences were central to the narratives of women in our study, who were more likely to be unemployed and report severe food insecurity compared to the men. Previous studies among WLHIV across diverse settings have described loss of employment as a form of rejection and discrimination [11], including in the DR [24, 39]. Our findings confirm these and extend them to men living with HIV, albeit with the additional precarity and intensity for women, who, pre-diagnosis, were more likely to work in occupations that required contact with others, including at restaurants, doing domestic work, and providing childcare. These types of jobs make WLHIV even more vulnerable to job discrimination, given the close contact with people and misconceptions about risk through casual contact and the often informal nature of such positions. Campaigns to promote stigma reduction among employers and awareness raising among workers (e.g., about their rights) may be particularly needed in these sectors and among women.

Anticipated stigma was reflected in concerns about discrimination in hiring from both men and women in our study, who described refraining from even applying to jobs where they might be required to undergo HIV testing. Even though such testing is illegal in the DR, these laws have very little effect due to lack of monitoring and enforcement [18], highlighting the way institutional barriers are reinforced despite—and potentially masked by—surface legal protections. Previous studies in the DR have documented the negative impact of illegal HIV testing on mental health, employment, and engagement in HIV care among MSM and transgender women and have called for interventions to address these discriminatory employment practices [54]. Our study extends these findings of illegal HIV testing to women and men with HIV more generally, but also emphasizes how anticipated stigma that results from this practice negatively affects women's adherence to ART and engagement in care. Anticipated stigma has been shown to be associated with adverse health outcomes among WLHIV, including reluctance to seek HIV-related care [11]. Another study found that WLHIV in the DR avoided HIV clinics out of fear of having their status involuntarily disclosed if they are seen there [24]. Policy efforts to encourage universal testing and treatment strategies as a way to end the HIV epidemic must therefore also address HIV stigma; in the short run, considerations regarding to the organization and delivery of HIV care in such settings are needed to minimize involuntary disclosures.

Men and women also differed in their experiences of internalized stigma. Women described feelings of shame and guilt, and mental health challenges (including suicidality and depression), which sometimes prevented them from seeking treatment. These findings are consistent with previous studies with WLHIV across different settings in which women described similar feelings of guilt, stress, mental illness, and suicidality [11]. In the DR, WLHIV have particularly high levels of internalized stigma and depression [23]. In contrast, men did not describe (or described with less intensity) feelings of shame or negative self-concept. Some men exhibited more cavalier attitudes and attributed their HIV to being "in the street / en la calle", a phrase used to describe promiscuity and risky sexual behaviors, which may reflect dominant cultural norms that these behaviors are more socially accepted among men. A quantitative study found that Dominican men might adopt risky sexual behaviors to meet gendered expectations around masculinity, sexual prowess, and sexual performance (e.g., having multiple sex partners, consuming alcohol prior to sex, and sporadic condom use) [55]. Men in our study also commonly blamed women for their HIV-positive status, similar to findings from a study in Ghana, a dynamic that may cause more intensified isolation, stigma, and disempowerment among women [10].

Coping mechanisms and sources of resilience are critical in the management of HIV and men and women in our study described a variety of sources for social support, which enabled them to cope with marginalization they face. Sources of support included family and friends, the community (i.e., neighbors and church networks), institutional and government institutions, and other PLHIV. For women, emotional, moral, and financial support came from daughters, relatives, neighbors, and friends. Because men were less likely to disclose their HIV status to others, many did not describe social support obtained following a diagnosis–but were more likely to describe other PLHIV from their clinic to be a source of support. Further, more often than men, women described religiosity and spirituality as a source of strength, confirming quantitative findings of an earlier study in Ethiopia [56]. Some men did mention receiving institutional support such as food assistance, subsidized health insurance, and low-cost medications. Aside from monthly food assistance, other institutional supports were not commonly mentioned by women. Further, social support for both genders appeared to informal, rather than formal support groups, which have been previously described as important for reducing isolation and shame for women [11, 24]. A study involving WLHIV in the U.S. has described religiosity, spirituality, being in solidarity with individuals who share similar experiences, and social support as sources of resilience [8].

This study has several limitations. First, since we recruited individuals seeking care at an HIV clinic, the experiences may not reflect those of individuals less engaged in care. In addition, the inclusion of only two clinics may also be a limitation, as participants in other locations may face distinct challenges and experience stigma differently than participants engaged in the present study. Further, study clinics were located in urban or semi-urban regions, which may be associated with unique, place-based challenges and experiences that differ from those faced by individuals residing in rural communities. Our analysis compared findings between women and men who self-identified their gender as such; as such, we cannot speak to stigma experiences for other gender identities. In addition, we did not collect data on sexual orientation or involvement in sex work, which have been documented in the literature on stigma in the DR. Another limitation is that these findings may reflect a reluctance among men to discuss feelings in general (but particularly shame, guilt, mental illness) due to gendered cultural norms and expectations that do not encourage this emotional vulnerability among men.

Despite these limitations, this study is among the first studies of gendered differences in stigma between men and women living with HIV in the DR. Recruitment of a highly

marginalized group in a low-income setting provides valuable insight for development of targeted health promotion interventions to reduce stigma and improve HIV health outcomes.

The social, political, and economic conditions relevant to living with HIV in the DR underscore the intersectional nature of HIV-related stigma. The individual experiences of participants demonstrate their unique position in the macro social hierarchy, with each participant having multiple disadvantaged identities at the intersection of classism, sexism, and HIV-related stigma [57]. Ultimately, community-level and systems-level interventions are needed for all PLHIV (e.g., to prevent job discrimination and strengthen fragmented social networks and support), but WLHIV-specific interventions should consider the discrimination, violence, and stigma faced by women. Large scale educational and health promotion campaigns may also help reduce stigma by providing critical HIV-related knowledge about how HIV can be prevented, transmitted, and adequately treated. Further, interventions around gender-based violence and intimate partner violence are needed to both prevent instances of violence, case identification, and to provide support to victims. These initiatives may lend HIV more visibility and promote widespread awareness, normalizing and destigmatizing HIV and PLHIV, an effort that may reduce the impact of stigma and improve both HIV-related health outcomes and the overall quality of life of PLHIV.

## Supporting information

**S1 File. Inclusivity in global research.**
(DOCX)

## Acknowledgments

We thank the Dominican and Haitian participants who participated in the study, as well as clinic staff, especially the directors of the study clinics, Dr. Mirquella G. Rijo Ureña and Lcda. Chaira Rodríguez, and Dr. Johanny Tejada. We also thank Dr. Yarovis N. Ortíz Rivera and Dr. Eves Priscila Peyes Pujols, who provided data collection support as research interns with the Ministry of Public Health. Finally, we recognize the on-going assistance to ensure the study's success from Dr. Víctor Terrero, formerly Executive Director of CONAVIHSIDA and Dr. Ángel Díaz, Director of Research of the Facultad de Ciencias de la Salud (School of Health Sciences) at the Universidad Autónoma de Santo Domingo.

## Author Contributions

**Conceptualization:** Amarilis Then-Paulino, Kathryn P. Derose.

**Formal analysis:** Alane Celeste-Villalvir, Denise D. Payan, Gabriela Armenta, Kartika Palar, Kathryn P. Derose.

**Funding acquisition:** Kathryn P. Derose.

**Investigation:** Ramón Acevedo, Maria Altagracia Fulcar.

**Methodology:** Denise D. Payan, Amarilis Then-Paulino, Ramón Acevedo, Maria Altagracia Fulcar, Kathryn P. Derose.

**Supervision:** Denise D. Payan, Amarilis Then-Paulino, Kathryn P. Derose.

**Writing – original draft:** Alane Celeste-Villalvir, Kathryn P. Derose.

**Writing – review & editing:** Denise D. Payan, Gabriela Armenta, Kartika Palar, Kathryn P. Derose.

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
