## [Decision Letter · Decision Letter 0]

30 Jan 2023

PONE-D-22-26942Exploring gender differences in HIV-related stigma and social support in a low-resource setting: A qualitative study in the Dominican RepublicPLOS ONE

Dear Dr. Derose,

Thank you for submitting your manuscript to PLOS ONE. After careful consideration, we feel that it has merit but does not fully meet PLOS ONE’s publication criteria as it currently stands. Therefore, we invite you to submit a revised version of the manuscript that addresses the points raised during the review process.

We look forward to receiving your revised manuscript.

Kind regards,

Adetayo Olorunlana, Ph.D.

Academic Editor

PLOS ONE

Journal Requirements:

The authors have no relevant financial or non-financial interests to disclose.

Reviewers' comments:

Reviewer's Responses to Questions

**Comments to the Author**

1. Is the manuscript technically sound, and do the data support the conclusions?

Reviewer #1: Yes

Reviewer #2: Partly

2. Has the statistical analysis been performed appropriately and rigorously? 

Reviewer #1: N/A

Reviewer #2: I Don't Know

3. Have the authors made all data underlying the findings in their manuscript fully available?

Reviewer #1: No

Reviewer #2: No

4. Is the manuscript presented in an intelligible fashion and written in standard English?

Reviewer #1: Yes

Reviewer #2: Yes

5. Review Comments to the Author

Reviewer #1: The authors describe that the consent form did not inform participants that the data could be made available to outside researchers and thus their IRB has informed them that they cannot make the data available outside of the study team.

Reviewer #2: OVERALL:

This is a robust paper that describes stigma among men and women living with HIV in the Dominican Republic. This manuscript is developed from secondary analysis of data from a parent study, where stigma was discussed indirectly. The authors conducted a content analysis based on pre-existing categories, guided by the HIV stigma framework. The authors describe similar and different ways that three types of stigma manifest in men vs. women. The manuscript adds to the HIV stigma literature by offering a gender-stratified lens, although it is unclear to this reviewer if this area of research is, in fact, previously unexplored.

This reviewer questions the overall soundness of the study findings based on questions of methodological rigor and analysis, highlighted by reporting most results in unsubstantiated quantitative terms. If the editors find this article suitable for their readership, I recommend a major revision with deep re-analysis to insure rigor, reporting and therefore updated discussion.

Background:

Authors provide helpful descriptions of various types of HIV related stigma and how stigma has been found to impact a range of socioecological levels of influence on peoples’ health and well-being. Identifying the HIV stigma Framework (lines 75-87) as a guiding conceptual framework is a strength of this paper.

The authors should revisit their presentation of the state of knowledge regarding the way HIV stigma has already been found to “be experienced differently by women compared to men”. (lines 88-97) This feels overstated, and can lead the reader to question why this analysis is being conducted in the first place, if the reasons for and prevalence of gender-based stigma are already known.

Last, the authors do make the case for comparing gendered experiences of HIV related stigma. However, they do not make the case (not even mentioned) for exploring experiences of social support and coping mechanisms. These factors need a rationale for why they are being studied. This is a theme throughout – it feels like social support and coping mechanisms were tacked on to the analytic plan without sufficient rationale or attention throughout.

Methods:

The authors claim to have analyzed the overall dataset ‘by gender’ but do not provide sufficient detail on their study design to support a gender-based analysis. First, more description is needed about recruitment (lines 157-8) of a ‘gender-balanced’ sample. How did you balance the sample? How many genders were included? How was gender identified to ensure the sample was ‘balanced’? did you exclude people of non-binary gender? In lines 172-3, gender options include ‘male, female’, which are labels for sex, not gender. This should be identified as a limitation, since it may have led to misclassification in the sample. Also, authors should indicate how they approached the gender analysis. Were the data analyzed separately by gender from the outset? Vs. Were the data analyzed as a full dataset and then segmented by gender? At what point did the analyses diverge?

I also have questions about the type of analysis that was conducted. From the authors’ descriptions, it appears that they in fact conducted ‘deductive’ content analysis, since they categorized their data by a pre-determined set of categories, according to categories within the HIV stigma framework ((ie experienced, anticipated, internalized) lines 189-200 ) and other a priori categories (ie social support).

In order to increase trustworthiness of the qualitative findings, I encourage the authors to follow COREQ guidelines to ensure they have included standard and sufficient info for the reader to determine trustworthiness in this manuscript. One example of a missing piece that should be included (when COREQ guidelines are followed) is description of any reflexive practices that occurred within the research team. This is just one example of what appears to be missing, but the article needs a thorough review to ensure other COREQ criteria are met minimally.

Results:

Where numbers are characterized (more, some, many, etc), please also include the numerator and denominator for each piece of information so that readers are clear of the proportions. This is crucial in qualitative work where it is not necessarily a given that data is self-initiated or prompted, or queried within the entire sample. It is noted that most of the results are reported in this enumerated fashion, which is not the intention of qualitative findings in general. Further, descriptions of one participant’s experiences are OK to describe, but it is preferable that the authors contextualize why they are exemplifying one participant rather than presenting the data as exemplary of a particular theme.

The analysis could be strengthened by deeper exploration of themes – there was lack of detailed thick description, which readers expect to see in qualitative reports. (p. 14&15, lines are not present) Instead the overwhelming reporting relies on quantitative comparisons of theme prevalence across men and women rather than identifying the qualitative characterization of men and women’s experiences. This makes the data read as potentially cherry-picked by the authors, and calls into question if there is additional, different, or more mature analysis to be done. I recommend conferring with a qualitative methodologist to examine these issues.

Table 1 –

Age is reported as a mean, but no standard deviation is reported as stated. Please make consistent. The ‘occupation’ category is confusing. These are not mutually exclusive categories as described For example, one can be self-employed, but only employed part-time. Can the authors clarify this? Additionally, please define Food Insecurity status.

Discussion:

Due to methodological shortcomings and questions over the robustness and trustworthiness of the analysis, I am refraining from commenting on the Discussion section.

6. PLOS authors have the option to publish the peer review history of their article (what does this mean?). If published, this will include your full peer review and any attached files.

Reviewer #1: No

Reviewer #2: No

---

## [Author Response · Author response to Decision Letter 0]

23 May 2023

See attached Response to Reviewers

---

## [Decision Letter · Decision Letter 1]

17 Jul 2023

PONE-D-22-26942R1Exploring gender differences in HIV-related stigma and social support in a low-resource setting: A qualitative study in the Dominican RepublicPLOS ONE

Dear Dr. Derose,

Thank you for submitting your manuscript to PLOS ONE. After careful consideration, we feel that it has merit but does not fully meet PLOS ONE’s publication criteria as it currently stands. Therefore, we invite you to submit a revised version of the manuscript that addresses the points raised during the review process.

We look forward to receiving your revised manuscript.

Kind regards,

Adetayo Olorunlana, Ph.D.

Academic Editor

PLOS ONE

Journal Requirements:

Reviewers' comments:

Reviewer's Responses to Questions

**Comments to the Author**

1. If the authors have adequately addressed your comments raised in a previous round of review and you feel that this manuscript is now acceptable for publication, you may indicate that here to bypass the “Comments to the Author” section, enter your conflict of interest statement in the “Confidential to Editor” section, and submit your "Accept" recommendation.

Reviewer #1: All comments have been addressed

Reviewer #2: (No Response)

2. Is the manuscript technically sound, and do the data support the conclusions?

Reviewer #1: (No Response)

Reviewer #2: No

3. Has the statistical analysis been performed appropriately and rigorously? 

Reviewer #1: (No Response)

Reviewer #2: N/A

4. Have the authors made all data underlying the findings in their manuscript fully available?

Reviewer #1: (No Response)

Reviewer #2: No

5. Is the manuscript presented in an intelligible fashion and written in standard English?

Reviewer #1: (No Response)

Reviewer #2: Yes

6. Review Comments to the Author

Reviewer #1: (No Response)

Reviewer #2: This article has been much improved since the revision and most reviewer comments have been addressed, although there is one outstanding concern that is a critical one:

My previous review highlighted concerns about the trustworthiness of the qualitative analysis, because of key issues with the methods. The authors attempted to address my concerns by adding more quotes and providing more explanation for their analytic approach. While their changes did improve the robustness of their analysis, it did not address my biggest concern about the trustworthiness due to methodological shortcomings. If anything, their response conjured more concern about methodological appropriateness of the analytic method they used and, subsequently, how they report their findings.

In the previous and current manuscript versions, the authors state they conducted content analysis. First, it is not clear to me the rationale for using content analysis, since their research question does not fit squarely in the more typical types of research questions where content analysis is used (i.e. how topics are communicated/discussed, presence or absence of a theme, etc). I could imagine a choice to use content analysis being made so the researchers could determine, for example, differences in theme presence or salience between the 2 gender groups. However, the authors specifically chose not to include any quantified analysis (no numerators/denominators) as I had offered as a possibility in my last review, and they do not explain how the meaning units/codes were determined and related to one another. It is not essential to present quantitative data here, but it is critical in content analysis that the data segments that were coded into meaningful units are in the very least well defined and described. The analytic process required for this is not presented, and therefore I assume was not conducted. Second, they omit important descriptions of the content analysis approach they did take, if in fact content analysis was used. They do not describe if it was a latent vs. semantic content analysis; conceptual vs. relational , etc.. Third, the references for their methodological approach mainly refer to grounded theory rather than content analysis. So it leaves me further questioning the overarching methodology and subsequent findings, if in fact they conducted modified grounded theory, thematic analysis or some other approach. While possible that the term ‘content analysis’ is merely being misused or the methods are just not adequately described, either way this is a critical flaw in this manuscript and indicates the need for an additional or different analysis to be conducted.

To start, here is a good article that helps describe the difference bw grounded theory/content analysis. I think articles like these (and/or qual methods texts) are helpful in understanding nuances in qual methods approaches:

Cho JY, Lee EH. Reducing confusion about grounded theory and qualitative content analysis: Similarities and differences. Qualitative report. 2014 Aug 11;19(32).

7. PLOS authors have the option to publish the peer review history of their article (what does this mean?). If published, this will include your full peer review and any attached files.

Reviewer #1: **Yes: **Noelle R Leonard, Ph.D.

Reviewer #2: No

---

## [Author Response · Author response to Decision Letter 1]

1 Aug 2023

See attached Response to Reviewers document

---

## [Editor Report · Decision Letter 2]

7 Aug 2023

Exploring gender differences in HIV-related stigma and social support in a low-resource setting: A qualitative study in the Dominican Republic

PONE-D-22-26942R2

Dear Dr. Derose,

We’re pleased to inform you that your manuscript has been judged scientifically suitable for publication and will be formally accepted for publication once it meets all outstanding technical requirements.

Kind regards,

Adetayo Olorunlana, Ph.D.

Academic Editor

PLOS ONE
---

## [Editor Report · Acceptance letter]

15 Aug 2023

PONE-D-22-26942R2 

Exploring gender differences in HIV-related stigma and social support in a low-resource setting: A qualitative study in the Dominican Republic 

Dear Dr. Derose:

I'm pleased to inform you that your manuscript has been deemed suitable for publication in PLOS ONE. Congratulations! Your manuscript is now with our production department. 

Kind regards, 

on behalf of

Associate Professor Adetayo Olorunlana 

Academic Editor

PLOS ONE